# Genetic Ancestry and Self-Reported “Skin Color/Race” in the Urban Admixed Population of São Paulo City, Brazil

**DOI:** 10.3390/genes15070917

**Published:** 2024-07-13

**Authors:** Jaqueline L. Pereira, Camila A. de Souza, Jennyfer E. M. Neyra, Jean M. R. S. Leite, Andressa Cerqueira, Regina C. Mingroni-Netto, Julia M. P. Soler, Marcelo M. Rogero, Flavia M. Sarti, Regina M. Fisberg

**Affiliations:** 1Department of Nutrition, School of Public Health, University of São Paulo, São Paulo 01246-904, Brazil; jaquelinelopes@usp.br (J.L.P.); jeanswb@usp.br (J.M.R.S.L.); mmrogero@usp.br (M.M.R.); 2Department of Statistics, Institute of Mathematics and Statistics, University of São Paulo, São Paulo 05508-090, Brazil; milaalvesdesouza@gmail.com (C.A.d.S.); jmontoya@usp.br (J.E.M.N.); pavan@ime.usp.br (J.M.P.S.); 3Department of Statistics, Federal University of Sao Carlos, São Carlos 13565-905, Brazil; acerqueira@ufscar.br; 4Human Genome and Stem Cell Research Center, Department of Genetics and Evolutionary Biology, Biosciences Institute, University of São Paulo, São Paulo 05508-090, Brazil; renetto@ib.usp.br; 5School of Arts, Sciences and Humanities, University of Sao Paulo, São Paulo 03828-000, Brazil; flamori@usp.br

**Keywords:** human population genetics, diversity, genomics, ethnicity, multi-ancestry, racial identity, Brazilian population, single-nucleotide polymorphisms (SNPs), genome-wide analysis, ancestry markers, health survey

## Abstract

Epidemiological studies frequently classify groups based on phenotypes like self-reported skin color/race, which inaccurately represent genetic ancestry and may lead to misclassification, particularly among individuals of multiracial backgrounds. This study aimed to characterize both global and local genome-wide genetic ancestries and to assess their relationship with self-reported skin color/race in an admixed population of Sao Paulo city. We analyzed 226,346 single-nucleotide polymorphisms from 841 individuals participating in the population-based ISA-Nutrition study. Our findings confirmed the admixed nature of the population, demonstrating substantial European, significant Sub-Saharan African, and minor Native American ancestries, irrespective of skin color. A correlation was observed between global genetic ancestry and self-reported color-race, which was more evident in the extreme proportions of African and European ancestries. Individuals with higher African ancestry tended to identify as Black, those with higher European ancestry tended to identify as White, and individuals with higher Native American ancestry were more likely to self-identify as Mixed, a group with diverse ancestral compositions. However, at the individual level, this correlation was notably weak, and no deviations were observed for specific regions throughout the individual’s genome. Our findings emphasize the significance of accurately defining and thoroughly analyzing race and ancestry, especially within admixed populations.

## 1. Introduction

The terms race, ethnicity, and ancestry have been widely used interchangeably in population studies [1], highlighting a discrepancy in how society defines, labels, and categorizes individuals. This inconsistency extends to how populations are classified for research purposes and how findings are applied to promote health for all human beings independent of their pre-defined classification [2].

In the last three decades, discussions surrounding these three terms have significantly intensified in population health and medical studies as a response to the rising debates concerning the importance of the ethnic–racial aspects in health–disease processes [3], and after a series of circumstances of racial injustice stemming from the utilization of race and ethnicity as biologic constructs to foster hierarchical discrimination [4].

Efforts have been made to set recommendations regarding the use of these terms to enhance their application in research; however, no established consensus exists [5]. In general, most definitions consider that race and ethnicity are dynamic social constructs influenced by geographic, cultural, and sociopolitical factors [1,6,7]. However, in recent years, the term race has been considered pejorative because it classifies humans based on phenotype—observable physical traits—perpetuating the incorrect belief in biological differences among sociopolitically constructed racial categories [1,8], whereas studies of human genetics have consistently shown that the concept of race has no genetic or scientific basis [1,8,9]. Many authors recommend using the terms ethnic group or ethnicity instead of race, as these terms encompass shared cultural background, including nationality, language, religion, and dietary practices, and may also, but not necessarily, reflect common biological characteristics [1,6]. Conversely, the term ancestry can be defined geographically (ancestors from similar regions), genealogically (an individual’s ancestral pedigree), or genetically [1,6,7].

Understanding genetic ancestry provides an opportunity for a deeper comprehension of human history, movement, evolution, and admixture. Likewise, distinct ancestry backgrounds result in different allelic frequencies, potentially influencing aspects such as health, longevity, disease susceptibility, and severity [10], response to drugs [11], and accuracy of genetic risk scores [12].

Nevertheless, epidemiological studies frequently categorize groups based on phenotypes that inadequately represent genetic ancestry [13], relying on self-reported race, skin color, or ethnicity to determine an individual’s ancestral origin, a subjective and arbitrary classification. This approach may lead to the inclusion of individuals with vastly different levels of population ancestry in the same group, particularly among those of multiracial backgrounds, making the categorization challenging [2,14].

The Brazilian population has one of the most heterogeneous genetic constitutions in the world with an extensive recent admixture, the result of nearly 500 years of interaction between three primary ancestral roots, Sub-Saharan African, European, and Native American populations, reflecting its history of colonization, slavery, and migration [3,15]. These three populations present different proportions of ancestries when comparing Brazilians with other admixed populations with similar histories of colonization from Latin America and the Caribbean [16,17,18], as well as among Brazilians from different geographical regions of the country [15,19,20,21,22,23].

Many epidemiological studies with the Brazilian population use the census classification [13], based on five ethnoracial self-classification groups [24], Black (”Preta”), Indigenous (“Indígena”), Mixed (“Parda”), White (“Branca”), or Yellow (“Amarela”), where the skin tone is the primary defining characteristic for clustering individuals. Nonetheless, previous studies have indicated that in Brazil, skin color is a poor predictor of genetic ancestry [25].

In this context, exploring the genetic ancestry of the study population from the Health Survey of São Paulo with a Focus on Nutrition Study (2015 ISA-Nutrition) is the first step to evaluate how genetics is associated with lifestyle, environmental, and biochemical factors in the development of cardiometabolic diseases in the population of Sao Paulo, the largest Brazilian city. The proposed genetic analysis and data analysis workflow can contribute to the basis of genetics-based medical therapies. In addition, investigating the interrelation between genetic ancestry and ethnoracial self-classification may contribute to a more comprehensive understanding of human genetic diversity and self-perceptions of identity in a multiethnic society.

Therefore, this study aimed to characterize both the global and the local genome-wide genetic ancestries and to assess their relationship with self-reported skin color/race in an admixed representative urban population of Sao Paulo city.

## 2. Materials and Methods

### 2.1. Study Design, Population, and Skin Color/Race Information

We analyzed 841 individuals living in permanent private households from the cross-sectional, population-based Health Survey of São Paulo with a Focus on Nutrition Study (2015 ISA-Nutrition), which aims to evaluate the relationship between lifestyle, environmental, biochemical, and genetic factors in the development of cardiometabolic disease in the population of São Paulo city. Details of the study were described elsewhere [26].

Briefly, 2015 ISA-Nutrition is a sub-sample of the Health Survey of São Paulo (ISA-Capital), which evaluated the health status and the use of health services of the population. The study assessed a probabilistic sample of individuals aged 12 years and older, not pregnant or lactating, and living in permanent households from the urban area of São Paulo city, Southern Brazil, with sampling stratified by clusters in two stages (urban census tracts and households) to ensure representativeness at the population level. In ISA-Capital, demographic and other information were collected in the households throughout the year 2015 using a structured questionnaire applied by trained interviewers to 4024 individuals [27]. In this interview, among other questions, participants were asked the following: “Which is your skin color or race?”. Therefore, skin color/race was self-reported by study participants, categorized by themselves according to the five ethnoracial categories used by the Brazilian census [24], which relies on self-perception of skin pigmentation: Black, Indigenous, Mixed, White, or Yellow. Alternatively, they could answer “other”, and specify any description they prefer for their skin color/race. Other terms used to describe the admixed character of the Brazilian population (e.g., “moreno/a”, “moreno/a claro/a”, and “moreninho/a”, which correspond to terms referring to tanned/light dark skin color) were collapsed into the category Mixed (n = 36) [28].

### 2.2. Genetic Information and Quality Control (QC)

In 2015 ISA-Nutrition, blood samples, and other clinical measurements were collected by a trained nurse instructed to follow the standardized procedures for collecting biological samples from 901 individuals from February 2015 to February 2016 during a visit to the households. Details of data collection and processing were previously described [26]. Genomic DNA was obtained from peripheral blood samples with EDTA stored at −80 °C, which was thawed and submitted to an automated extraction protocol in the QIAsymphony SP BioRobot with the QIAsymphony DNA Midi Kit 96 (Qiagen, Hilden, Germany), in accordance with the manufacturer’s instructions, at the Human Genome and Stem Cell Research Center of the University of São Paulo (IB-USP). For 40 samples with less than 1 mL of blood, the automated extraction was unfeasible; therefore, a protocol for salting out extraction was used [29]. One sample with less than the necessary volume for the genotyping protocol was excluded from the analysis.

The quantification and quality of DNA samples were evaluated with the NanoDrop™ 2000 spectrophotometer (Thermo Fisher Scientific, Waltham, MA, USA). Approximately 130 samples did not present acceptable purity ratios and were subjected to the purification process (Autopure LS-Qiagen, Hilden, Germany). Once all samples evaluated via spectrometry were within the quality standards in relation to the expected purity ratios, quantification via fluorescence was performed using Qubit™ dsDNA BR DNA Quantification Kit in the Qubit^®^ 2.0 fluorometer (Thermo Fisher Scientific, Waltham, MA, USA). Additionally, a DNA integrity check was performed on randomly selected samples from each batch submitted to automated or manual extraction based on electrophoresis.

Genotype calling was performed for 846 samples using Axiom™ 2.0 Precision Medicine Research Array in the Thermo Fisher Scientific Laboratory (Affymetrix Inc, Santa Clara, CA, USA). The selection of 846 samples to be analyzed in nine 96-array plates (n = 864 minus 18 for control, as per manufacturer’s guidance) was randomized, prioritizing individuals with complete anthropometric and laboratory information as well as unrelated individuals. SNP quality control (QC) was initially performed using Affymetrix Power Tools (APT 1.16.0) according to the Affymetrix best practices QC criteria, in which 873,177 markers are recommended from an initial panel of 920,745 markers, considering a call rate <98% besides other criteria (Axiom™ Genotyping Solution Data Analysis User Guide) [30].

The initial sample assessment of the processing quality was performed using Axiom™ Analysis Suite software v.4.0, and five individuals were excluded due to a call rate <0.95. Therefore, the sample for the analysis comprised 841 individuals, including 423 males (50.3%) and 418 females (49.7%), with a mean age of 43.8 years, ranging from 12 to 93 years, distributed in 629 different households. For the ancestry analyses, kinship was addressed, considering that the sample contained substantial familial relatedness, as expected from the household sampling design. In each household, self-reported kinship with the head of the household was assessed and used as a basis for selecting genetically independent individuals. For example, if individuals were spouses sharing the same household, they were considered unlikely to have a genetic relationship. Using the 707 pseudo-independent individuals, quality control was performed, and SNPs with a minor allele frequency (MAF) <0.05 or a Hardy–Weinberg equilibrium *p*-value of less than 10^−6^ were excluded. Additionally, SNPs in linkage disequilibrium (LD) (window size = 50 SNPs, shift step = 5 SNPs, and r^2^ = 0.5) were removed, and the genomic relatedness matrix (GRM) was computed with the remaining SNPs for all 707 individuals. These SNPs were used to estimate identical-by-descent and GRM estimations >0.125, which is the expected genomic relatedness for second-degree and any closer pair of relatives (2K = 1/22 = 1/4 = 0.25 and K = 0.125). Therefore, from the 841 individuals, 121 were excluded from the global ancestry analysis. A study sample flowchart with details about the analyses performed is presented in Appendix A.

To conduct the global ancestry analysis, SNP quality control (QC) was performed in PLINK 2.0 using 873,177 recommended SNPs, excluding markers without rs notation located in sexual chromosomes, HLA-Palindromics, and high-linkage-disequilibrium (LD) regions, SNPs with a minor allele frequency (MAF) <10^−4^, and those with extreme deviation from the Hardy–Weinberg equilibrium (*p* < 10^−6^), resulting in 711,087 SNPs. Additionally, these SNPs were thinned to reduce the residual LD (window size = 50 SNPs, shift step = 5 SNPs, and r^2^  =  0.2), resulting in 417,192 high-performing SNPs.

In the analyses of population structure and global ancestry inference, the 2015 ISA-Nutrition sample (ISA) was compared with 1000 Genomes Project phase 3 (1KGP) [31] (GRCh37/hg19 assembly; ftp://ftp.1000genomes.ebi.ac.uk/vol1/ftp/release/20130502/ (accessed on 8 March 2023)), which is a public dataset with common human genetic data from self-reported healthy participants. Study participants of Sub-Saharan African (AFR) ancestry were selected from the following groups: African Caribbean in Barbados (ACB), African Ancestry in Southwest US (ASW), Esan in Nigeria (ESN), Gambian in Western Division (GWD), Luhya in Webuye (LWK), Mende in Sierra Leone (MSL) and Yoruba in Ibadan (YRI). Native American (AMR) individuals were selected from the following groups: Colombian (CLM), Mexican in Los Angeles (MXL), Peruvian in Lima (PEL), and Puerto Rican (PUR). East Asians (EAS) were selected from various populations, such as Chinese Dai in Xishuangbanna (CDX), Han Chinese in Beijing (CHB), Chinese in Denver, Colorado (CHD), Southern Han Chinese (CHS), Japanese in Tokyo (JPT), and Kinh in Ho Chi Minh City, Vietnam (KHV). Europeans (EUR) included Utah residents with Northern and Western European ancestry (CEU), Iberian populations in Spain (IBS), Finnish individuals in Finland (FIN), British individuals in England and Scotland (GBR), and Toscani individuals in Italy (TSI).

We selected 1585 individuals from a 1KGP sample with original ancestry >95% using FastStructure, which is an algorithm for population structure inference applicable to large genotype datasets (SNPs). This is based on a variational Bayesian inference approach that includes the variational expectation–maximization algorithm as its optimization algorithm [32,33]. We used the simple prior allele frequencies to explain the ancestry of the Sub-Saharan African, East Asian, Native American, and European superpopulations based on the known Brazilian demographic history [25,34]. Data from the South Asian population were tested but excluded from the analysis because of the extremely low representativeness in the population. The best conformation of k groups was executed to maximize the marginal probability and best explain the structure of the data, obtaining the best number of clusters k equal to 4. The output was processed in R to select samples with more homogeneous ancestry based on the proportion of ancestry of individuals in each superpopulation. In this way, the generated reference panel contained 557, 46, 504, and 478 homogeneous samples for the AFR, AMR, EAS, and EUR groups, respectively.

The 1KGP data were merged with the ISA dataset, removing 178,376 SNPs from ISA that did not match 1KGP. Multiallelic SNPs (with two or more variant alleles) were also excluded. Overall, 226,346 SNPs were evaluated. The number of markers excluded in each step is described in Appendix A, and the distribution of markers by chromosome before and after pruning is illustrated in Appendix A.

### 2.3. Population Structure and Global Ancestry Inference

The overall genetic ancestry, or global ancestry of an individual, is the relative proportion of the mosaic of ancestral blocks from each contributing ancestral population across the chromosome [35]. To evaluate the ISA population structure and to compare ISA and the four populations from 1KGP, principal component analysis (PCA) was performed using the *SNPRelate* package [36], in R software v4.1.0, with the computational support of the *gdsfmt* package, producing an environment optimized for high performance [37]. The genotype matrix, called G, has an NxP dimension, where N = 2305 individuals (ISA-Nutrition + 1KGP) and P = 226,346 markers were used in the analysis. The eigenvectors or principal components (PCs) were sorted in decreasing order of the corresponding eigenvalues, i.e., the first eigenvector (PC1) accounts for the largest variation in the data on G, the second eigenvector (PC2) accounts for the second largest, and so on. Global ancestry was estimated for the unrelated individuals (n = 720) and according to self-reported skin color/race (Black, Indigenous, Mixed, White, or Yellow).

GRM matrix, along with relatedness, was used to further adjust for population stratification in the estimation of global ancestry by applying the standardized loads of the pseudo-independent sample (n = 720) to the related sample (n = 121), resulting in estimates of global ancestry for the total ISA-Nutrition population (n = 841).

The individual ancestry proportion for the total ISA sample and according to self-reported skin color/race was estimated considering the proportions of AFR, AMR, and EUR ancestry, considering the extremely low proportion of EAS ancestry.

### 2.4. Local Ancestry Inference

Local ancestry is defined as the genetic ancestry of an individual at a particular chromosomal location, where an individual can have 0, 1, or 2 copies of an allele derived from each ancestral population [35]. Local ancestry was estimated using the RFMix v.1.5.4 software [38], considering the aforementioned reference panel, composed of 1585 individuals from 1KGP with an ancestry homogeneity of 95% [31]. The panel was previously phased using the SHAPEIT2 [39]. RFMix has a discriminative modeling approach that models the local ancestry in admixed populations and uses a conditional random field (CRF) parameterized by random forests trained on reference panels, improving the inference accuracy by the expectation–maximization (EM) iterative algorithm. The output documents from RFMix were formatted to create the ideograms using the *tidyverse* and *ggplot2* packages in R software v4.1.0.

### 2.5. Statistical Analyses

Absolute and relative frequencies were described for the ISA sample according to self-reported skin color/race and compared using either Pearson’s chi-square or Fisher’s exact tests where applicable (specifically for Black, Mixed, and White categories due to the notably low frequencies of Indigenous, Yellow, and unanswered responses). Medians and interquartile ranges were estimated to describe the distribution of ancestry proportions, considering their asymmetric distributions. Statistical significance was evaluated employing the Kruskal–Wallis rank test. The probabilities of individual self-reported skin color/race classification in relation to varying levels of Sub-Saharan African, Native American, and European ancestry was examined by describing the proportions of Black, Mixed, and White self-classification across ancestry quartiles and compared using quantile median regression models adjusted for age to estimate the strength of these associations.

## 3. Results

### 3.1. Population

The study population consisted of 841 individuals, of which 720 were unrelated. For the unrelated sample, the proportion of men was 52.9%, the median age was 50 years (IQR: 18–65), and the most frequent self-reported skin color/races were White (50.6%) and Mixed (36.3%) (Table 1). Similar results were observed for the whole sample (N = 841), with 50.3% being men, the median age being 46 years old (IQR: 18–64), and the most self-reported skin color/races being White (50.2%) and Mixed (36.7%) (Appendix A). The categories of self-reported skin color/race were balanced according to sex and different for age group, with younger individuals self-reporting a higher frequency of Black and Mixed skin color/race and older adults reporting a higher prevalence of White (Table 1 and Appendix A).

Considering that São Paulo is a cosmopolitan city, a destination for migrants from various parts of the country as well as from other nations, and that the geographical region of origin influences the ancestry profile, we evaluated the distribution of the individuals according to their place of birth (Appendix A). Most individuals were born either in São Paulo city (52%) or in other cities within São Paulo state (11%). Additionally, there were individuals who originated from all five geographical regions of Brazil, with the northeastern being the most prevalent (22%), alongside a small percentage originating from other countries (3%).

### 3.2. Global Ancestry and Self-Reported Skin Color/Race

According to the evaluation of global genetic ancestry, the median ancestry estimates were as follows: 71.5% European descent, 18.2% Sub-Saharan African, and 6.1% Native American descent (69.7%, 18.6%, and 6.9%, respectively, for the total population). Those who self-reported as Black had 56.1% Sub-Saharan African, 35.8% European, and 5.6% Native American ancestries; those who self-reported as Mixed presented median ancestries of 62.3% EUR, 26.5% AFR, and 8.5% AMR; and those who self-reported as White had a median of 86.3% EUR, 7.4% AFR, and 3.6% AMR ancestries (Figure 1 and Table 1). The East Asian ancestry and the self-reported Indigenous, Yellow, and Not Answered categories are not presented in Figure 1 due to the very small proportion of these in the population.

Most of the population (80.3%) exhibited predominant European ancestry (>50%), which was higher for individuals who self-reported White (94.8%), lower for Mixed (80.5%), and even lower, but still important, for Black (18.9%). Nearly all individuals evaluated presented some degree of European ancestry (>0%). African ancestry was predominant (>50%) in 8.5% of the population, present in 56.8% of individuals who self-reported Black skin color/race and in 6.9% of those with Mixed skin color/race. Eighty-two percent of the population presented African ancestry in some degree, being present in 100% of those who self-identified as Black, 97.3% as Mixed, and 67% as White. Native American ancestry was not predominant in the population; however, 80.7% of individuals presented it to some degree (Table 1).

Self-reported skin color/race (Black, Mixed, and White) according to quartiles of individual Sub-Saharan African, Native American, and European ancestry are presented in Table 2 for unrelated samples and Appendix A for the total sample. A higher likelihood of self-reporting as Black was observed among individuals in the highest quartile of African ancestry, and a higher probability of self-reporting as White among the highest quartile of European ancestry. Those above the median of Native American ancestry were more likely to be self-reported as Mixed compared to Black. A small percentage (5.2%) of those identifying as White were found in the highest quartile of African ancestry, and 18.9% of those who self-reported as Black and 17.6% as White were situated at the highest quartile of Native American ancestry, whereas 6.9% of those who self-reported as Mixed were at the highest quartile of European ancestry.

The results derived from the quantile regression analysis showed varying degrees of association between self-reported skin color/race and genetic ancestry. Notably, the pseudo R^2^ values indicated the variance explained by the model of 0.35 for African ancestry, 0.07 for Native American ancestry, and 0.30 for European ancestry in the analysis comparing individuals above or below the median of genetic ancestry. Similar results were observed for the 0.75 model (0.36, 0.02, and 0.34, respectively).

In the determination of the population structure using PCA, the first and second components explained 35.8% of the genetic variation in the ISA-Nutrition sample and 84.8% in ISA together with the selected sample from 1KGP (Appendix A). The correlation of the first and second principal components (PCs) per chromosome in the ISA sample and ISA + 1KGP is uniform, as shown in Appendix A. The correlation between the first six principal components in the PCA analysis is plotted in Appendix A.

The results of the genome-wide PCA of the 720 ISA samples using four race/ethnic groups from 1KGP as reference samples (Sub-Saharan African = AFR, Native American = AMR, East Asian = EAS, European = EUR) are presented in Figure 2a. The results are projected onto the first two axes of maximal genetic variation (PC1 × PC2). As an admixed population, the individuals from ISA presented an accentuated internal dispersion. They were spread among the 1KGP populations, although most individuals were distributed between Europeans and Africans rather than towards Native Americans or East Asians.

The results are also presented according to the self-reported skin color/race (Figure 2b). In general, people who self-reported their skin color/race as White are closer to the European 1KGP population, despite some observations of a spread toward Native Americans and Africans. Those who self-reported as Black are distributed closer to Africans, but most of them not exactly in the same place in which the 1KGP AFR population is displayed spread towards EUR. Those who self-reported as Mixed are spread mostly between Europeans and Africans, despite some of them being spread in the direction of Native Americans. The position in the graph of individuals who self-reported as Yellow is divided between EAS and AMR. The same results of Figure 2 are presented in Appendix A, with the skin colors displayed separately for each category for easier visualization of the results.

The individual ancestral proportions of Sub-Saharan African, Native American, and European 1KGP populations, based on values obtained during PC analysis, were estimated for the total ISA population and according to the most frequent self-reported skin color/races: Black, Mixed, and White (Figure 3). In the ISA population, the component with the highest proportion was the most frequent in Europeans, followed by the components most frequent in Africans and Native Americans. Among those who self-reported their skin color/race as White, Europeans were the most frequent, despite the important proportion of the other two populations in this group. There were some cases of individuals who self-reported White without proportions of European ancestry. Those who self-reported as Mixed presented the highest mixture of components from the three populations compared to the other skin color/race groups, yet the most prevalent was European. The frequency of individuals who self-reported as Black was smaller than that of the White and Mixed groups. Yet, the number was sufficient for the analysis, which showed that these individuals had a higher representation of African ancestry despite also presenting an admixture pattern.

### 3.3. Local Ancestry and Self-Reported Skin Color/Race

Figure 4 presents the mean proportion of genetic local ancestry for each chromosome for AMR, AFR, EAS, and EUR ancestries. The proportions were homogeneously distributed throughout the genome, with a higher mean proportion of EUR ancestry, followed by AFR ancestry, a small proportion of AMR ancestry, and a very small proportion of EAS, without deviations for specific regions, as shown in Figure 5.

We scanned the genomic profile for signs of positive selection in the Brazilian population when analyzing the mean proportion of local ancestry for each genome variant in each of the four ancestral groups, and no deviations greater than 4.42 SD were found from the average. We adopted a significant selection measure deviation greater than 4.42 SD, since it is equivalent to a *p*-value < 10^−5^, as demonstrated in literature via neutrality simulation tests [22,40].

The average proportions of local ancestry in all samples and all SNPs were 0.22 (SD = 0.011 between SNPs) for the Sub-Saharan African, 0.051 (SD = 0.028) for the Native American, 0.022 (SD = 0.022) for the East Asian, and 0.70 (SD = 0.024) for the European population as ancestral reference.

The individual local ancestry is represented for each subject in the sample using ideograms, with examples according to self-reported skin color/race illustrated in Figure 6. A very different pattern of ancestries was observed among individuals, including those who had the same skin color/race self-classification.

## 4. Discussion

This study performed a population-based genome-wide analysis involving individuals from São Paulo city to explore both global and local genetic ancestry, along with its association with self-reported skin color/race. Our findings corroborated the admixed nature of the population, delineating its ancestry into three primary parental populations [3,15], irrespective of skin color classification: substantial European ancestry, a significant degree of Sub-Saharan African ancestry, and a uniformly small degree of Native American ancestry. We observed a correlation between global genetic ancestry and self-reported skin color/race. Individuals with elevated African ancestry were more likely to self-report as Black, while those with heightened European ancestry were more likely to self-identify as White. Furthermore, individuals with higher Native American ancestry presented a greater tendency to self-report as Mixed. However, no deviations were observed for specific regions throughout the individual’s genome, and the correlation between skin color/race and genetic ancestry was notably weak at the individual level. This is evident as some individuals self-reported as Black with less than 35% African ancestry, while others self-identified as White despite having significant African ancestry or predominantly East Asian or Native American local ancestries.

Brazil has a large territory, and different population groups have migrated to different parts of the country throughout history. Consequently, significant differences in genetic ancestry may be found depending on the specific geographic region of Brazil being investigated [15]. In the present study, with adolescents and adults living in São Paulo city, the global proportion of the three main ancestral roots (71.5% European, 18.2% African, and 6.1% Native American) was similar to that observed in other studies with individuals from the southeastern region (N = 264), 76.9% EUR, 13.8% AFR, and 7.0% AMR [22], from Sao Paulo state (N = 4338), 72.5% EUR, 18.1% AFR, and 8.1% AMR [15], and from older adults from São Paulo city (N = 1171), 72.6% EUR, 17.8% AFR, and 6.7% AMR [23], despite the methodological differences in assessing genetic ancestry across the studies.

Research conducted in other regions of Brazil has revealed differing proportions of ancestry. For instance, in Salvador city, located in the northeastern region, proportions were identified as 42.9% EUR, 50.8% AFR, and 6.4% AMR (N = 1309) [21], and in Belém city, situated in the northern region, the proportions were notably distinct, with figures showing 55.5% EUR, 17.6% AFR, and 26.9% AMR (N = 1044) [15]. Differences according to genetic ancestry methods and population selected may also influence these proportions, e.g., the ancestry proportions in Ilhéus (city from Bahia, like Salvador) was 60.6% EUR, 30.3% AFR, and 9.1% AMR [41].

Despite the observed variations, the predominant European ancestral component remains consistent not only within the studied population but also across other geographical regions of Brazil. The median proportion of European ancestry in ISA was 71.5%, close to the 69.7, 60.6, 73.7, and 77.7% observed in the north, northeast, southeast, and southern regions, respectively [25]. One plausible explanation has been denominated as the “Whitening of Brazil”, which attributes this framework to a substantial influx of millions of immigrants from Europe and the Middle East, particularly during the period from 1872 to 1975 [25,42]. This configuration is similar to findings from Puerto Rico (73.2% European, 13.9% African, and 12.9% Native ancestries) and different to those from other Latin American countries with population data available in the 1000 Genomes Project, such as Peru (20.2% European, 2.5% African, and 77.3% Native ancestries) or Mexico (48.7% European, 4.3% African, and 47.0% Native ancestries) [43], illustrating that while genomes from Latin American populations are a mosaic of the three ancestral populations, the proportion of each varies notably among countries [18].

The distribution of self-reported skin color/race among ISA population, classified based on their self-perception of skin color using the five ethnoracial categories outlined by the Brazilian census [24], closely resembles the official statistics of the southeastern Brazilian population, 11.2% Black, 37.3% Mixed, and 50.1% White [44], compared with 10.3%, 36.3%, and 50.6%, respectively, observed within the ISA sample. Over the past decade, there has been an increase in individuals self-identifying as Black within the Brazilian population [44]. This could be attributed to advancements in race literacy, increased awareness through social movements, and the implementation of affirmative racial identity policies. This trend was observed in our study, with data collected in 2015, when younger individuals exhibited a higher prevalence of self-identifying as Black or Mixed in contrast to older adults, who reported a higher prevalence of identifying as White. This fact underscores the fluidity and dynamism of the concept of race/ethnicity as a dynamic social construct influenced by geographic, cultural, and sociopolitical factors [1,6,7].

The diverse cultural semantic criteria used for skin color classification can also differ based on the population studied, given the substantial variability in the ancestry proportion within skin color/race categories across different regions of the country. A study evaluating the genomic ancestry of 934 Brazilians found differences in self-reported skin color/race across the country’s four major geographical regions [41]. Self-classified Black individuals exhibited substantial variations in European ancestry levels, ranging from 29.3% in Santa Catarina (South) to 53.9% in Bahia (Northeast). The median was 35.8% for the ISA sample. Some of the factors that can explain this difference are the culturally distinct concepts of race, as well as the effect of sunlight exposure on the skin in different regions [41]. Individuals self-classified as White had predominant European ancestry, ranging from 66.8% in Bahia (Northeast) to 86.1% in Rio de Janeiro (Southeast). A slightly higher median (86.3%) was observed in ISA. In addition, the most evident diversity in genetic ancestry proportions was observed among self-classified Mixed individuals: in Pará (North), the average European ancestry was 68.6%, followed by 20.9% Native American ancestry and 10.6% African ancestry, in contrast to 44.2% European, 11.4% Native American and 44.4% African ancestry in Rio Grande do Sul. In the current study, those who self-reported as Mixed presented a median of 62.3% European, 26.5% African, and 8.5% Native American ancestry. These findings reinforce the complexity of the multiethnic genetic framework encompassing the term “Mixed” within the Brazilian population [34].

To explore the correlation between ethnoracial classification and genomic ancestry, we employed three distinct analytical approaches. First, graphical analysis was utilized to visually represent the distribution of individuals based on the median proportion of ancestries, in relation to the reference populations from 1KGP, and the individual proportions of ancestries according to self-reported skin color/race. These results consistently demonstrate the substantial admixture of the studied population, as well as the higher proportion of European ancestry among those who self-reported as White, higher proportion of Sub-Saharan African ancestry among those who self-reported as Black, and intermediate proportions of these ancestries among those who self-reported as Mixed, despite not being a unanimous pattern among all individuals. These findings are aligned with those of previous studies with Brazilian samples [20,22,23].

Second, the evaluation of quartiles of ancestries compared using quantile regression models confirmed the significant association between global genetic ancestry and self-reported skin color/race, and indicated that it was more consistent in the extremes of the high and low proportion of African and European ancestries. However, the models were unable to accurately represent the heterogeneity and proportions of individuals’ genetic ancestry, considering the substantial percentage of variation in the models that remained unexplained by self-reported ethnoracial groups. For instance, the r^2^ value for the African ancestry model was 0.35, which was lower than the r^2^ value of 0.63 observed in a study of older adults from São Paulo [23] and the r^2^ value of 0.50 in the city of Pelotas (South) [20]. Conversely, it was higher than the r^2^ values of 0.22 and 0.13 observed in Bambuí (Southeast) and Salvador (Northeast), respectively [20].

To refine our understanding, the third analytical approach was local ancestry inference. Patterns of genetic ancestry have been used to detect post-admixture selection. One commonly used method involves identifying genome regions exhibiting local ancestry outliers when compared with the ancestry distribution across the genome, considering each locus independently. A recent study with Brazilians observed decreased European ancestry tracts on chromosome 8p23.1, followed by an excess of Native American ancestry tracts in this same region [45]. Examining the distribution of the average local ancestral proportion via SNPs for each ancestral reference group, our study revealed no indication of genomic selection within our Brazilian admixed population. However, it is important to acknowledge the potential existence of false negatives and highlight the lack of theoretical models specifically designed to evaluate expected ancestry distributions representative of diverse demographic scenarios, particularly those pertinent to our Brazilian population [40,46]. In addition, the average genome-wide local ancestry exhibited congruent findings when compared to the global ancestry inference within our sample, suggesting consistency between the two methods of ancestry inference.

The analysis of the local ancestry ideograms showed remarkable variability among individuals, including those self-classified in the same category. In agreement, a study with 1247 individuals from the United States observed 100% European ancestry among seven individuals who self-identified as African Americans [47], while another found a substantial admixture of African ancestry in European Americans [48]. These findings shed light on the potential bias that may arise when using race, ethnicity, and ancestry interchangeably. Each individual possesses a unique and distinct ancestry proportion. The imprecision and lack of inclusiveness in terminology and analysis can hinder the accurate characterization and translation of findings, thus restricting potential benefits while also contributing to possible damages, such as perpetuating racism [2,4].

Some limitations of this study should be acknowledged. First, the sample size in comparison to that of other genetic studies may be considered small. Nevertheless, this study contributes to the ongoing efforts aimed at addressing the marked under-representation of genomic data from Latin American countries, and to enhance diversity, a crucial step toward ensuring that genomic research can yield benefits for all [18]. Second, the use of publicly accessible reference populations does not necessarily reflect those that are directly ancestral to the studied population. Also, although the ancestry of human populations is a continuum, genetic ancestry information was simplified into continental ancestry classifications, imposing a potential simplification due to the notable similarity between continental ancestry categories and racial classifications [5]. However, this is inherent to the available methodology, and ongoing research utilizing new methodologies that may contribute to enhancing the understanding of population admixture.

## 5. Conclusions

In conclusion, despite the observed correlation between skin color/race and ancestry, this study underscores that they are not synonyms. It is not feasible to reliably predict the individual skin color/race solely based on their genetic ancestry proportion, vice versa. In our study, individuals self-identifying as Black exhibited significant European ancestry, while those self-identifying as White displayed varying degrees of African ancestry. Meanwhile, the category of individuals self-identifying as Mixed, constituting 36% of the studied population, encompassed a wide range of diverse ancestral compositions.

We emphasize that it is important to collect and analyze population group data in future epidemiological and clinical studies, especially considering that it is a key social classification for monitoring and addressing health inequities [8]. However, our findings highlight the critical importance of accurately defining these terms and thoroughly analyzing them, particularly within admixed populations, such as those in Brazil and other Latin American countries, as well as in North African populations, Near Eastern Asian or Eastern Mediterranean populations, where genetic analysis cannot always be performed and many people may identify with more than one race or ethnicity. Therefore, categories should not be considered absolute or viewed in isolation [6].

As precision medicine evolves and access to molecular testing expands, it is essential to address this issue adequately to improve the applicability of genetic research, particularly within underrepresented populations. To enhance study quality and improve overall health outcomes for mixed populations, regardless of skin color or ancestry, a comprehensive analysis of genetic markers should provide important information on health issues, such as disease susceptibility and treatment. In this context, evolutionary or personalized medicine could prove to be highly successful, ultimately improving health outcomes for the population as a whole.

## Figures and Tables

**Figure 1 genes-15-00917-f001:**
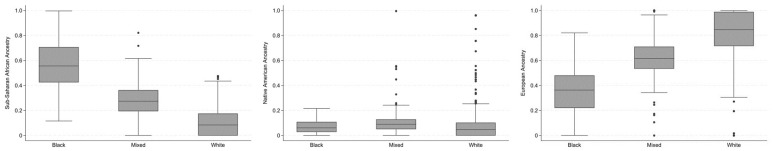
Box plot of median individual genetic Sub-Saharan African, European, and Native American ancestries according to self-reported skin color/race. The proportion of East Asian ancestry and the self-reported Indigenous, Yellow, and Not Answered categories are not presented due to the low frequency of observations. Detailed parameters (median, interquartile ranges, and *p*-values for statistical differences) are presented in Table 1.

**Figure 2 genes-15-00917-f002:**
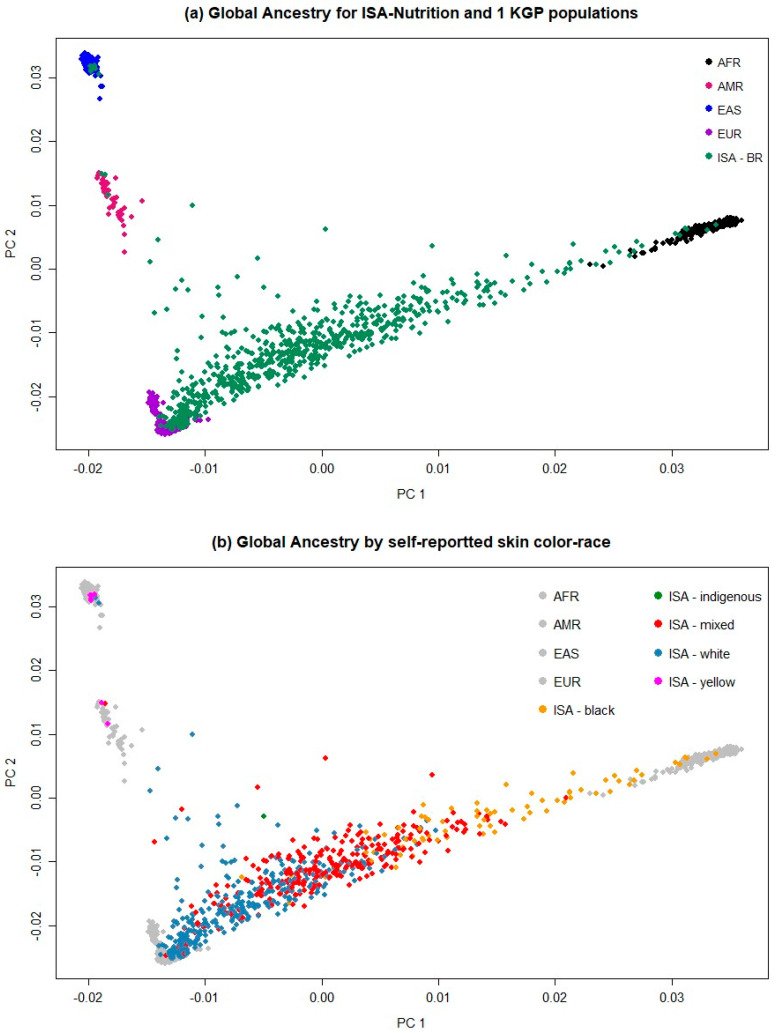
Scatterplot showing the first two principal components (PC 1 and PC 2) for the ISA-Nutrition sample (green), and Europeans (purple), Sub-Saharan Africans (black), Native Americans (magenta), and East Asians (blue) from 1KGP, for the total ISA-Nutrition population (**a**) and according to self-reported skin color/race, (**b**) Black (orange), Indigenous (green), Mixed (red), White (blue), or Yellow (pink). The same results are presented in Appendix A, with the self-reported skin color/race displayed separately for easier visualization of the results.

**Figure 3 genes-15-00917-f003:**
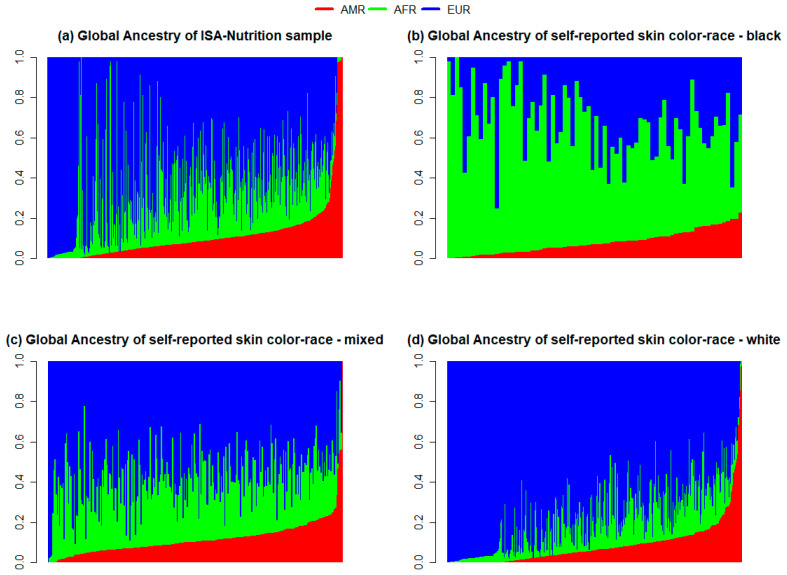
Global ancestry estimates obtained for total ISA population (**a**), and according to self-reported skin color/race: Black (**b**), Mixed (**c**), and White (**d**). Each column corresponds to one individual. Grades of blue, green, and red indicate European, Sub-Saharan African, and Native American ancestry components, respectively.

**Figure 4 genes-15-00917-f004:**
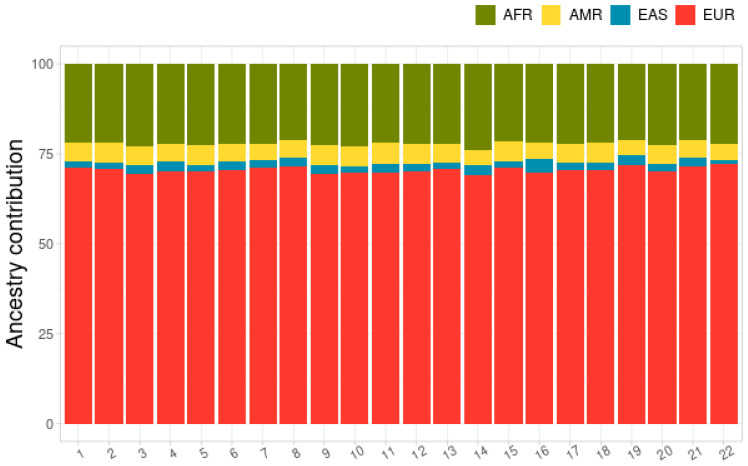
Proportion of genetic local ancestry for each chromosome for sub-Saharan African (AFR; in green), Native American (AMR; in Yellow), East Asian (EAS; in blue), and European (EUR; in red) ancestries.

**Figure 5 genes-15-00917-f005:**
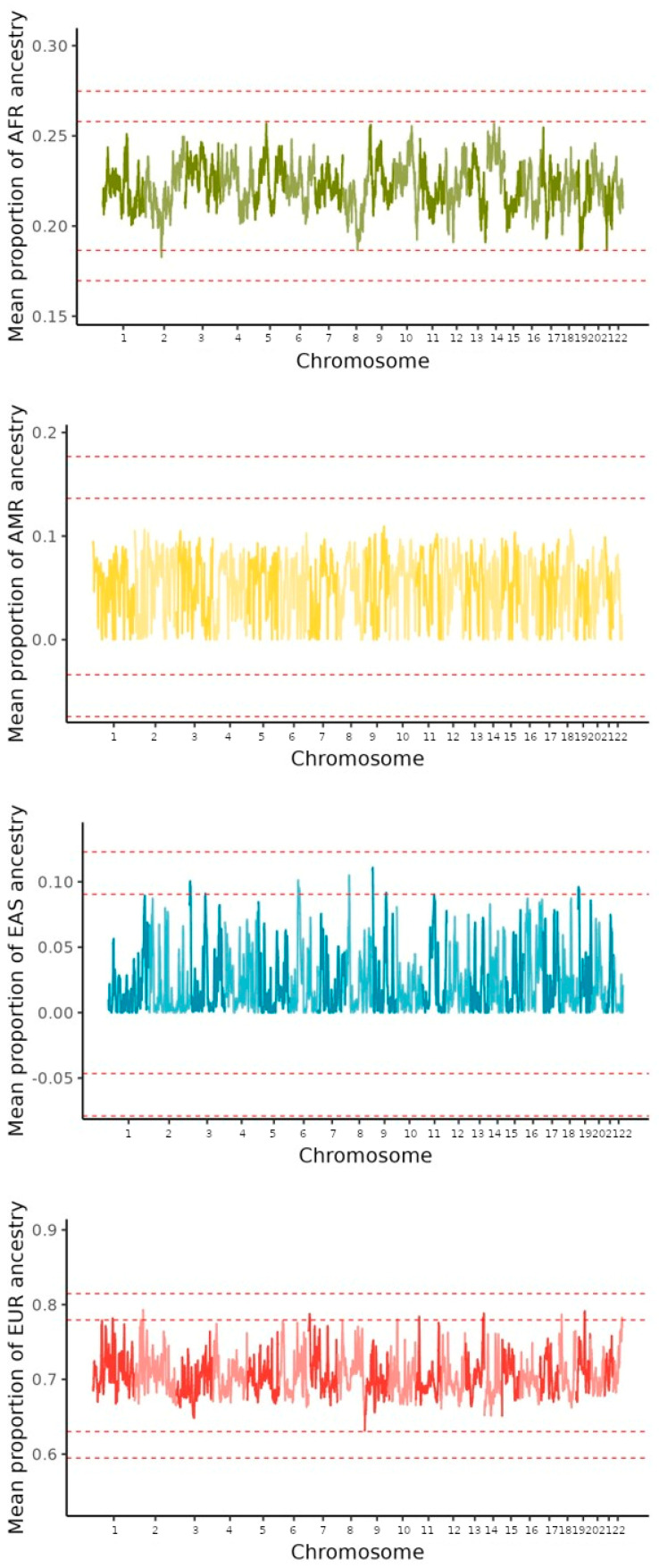
Dispersion of mean proportion ancestry across the genome estimated from Sub-Saharan African (AFR; in green), Native American (AMR, in Yellow), East Asian (EAS; in blue), and European (EUR; in red) reference panels. The internal dashed line indicates +/− 3 SD, and the external dashed lines indicates +/− 4.42 SD from the mean thresholds.

**Figure 6 genes-15-00917-f006:**
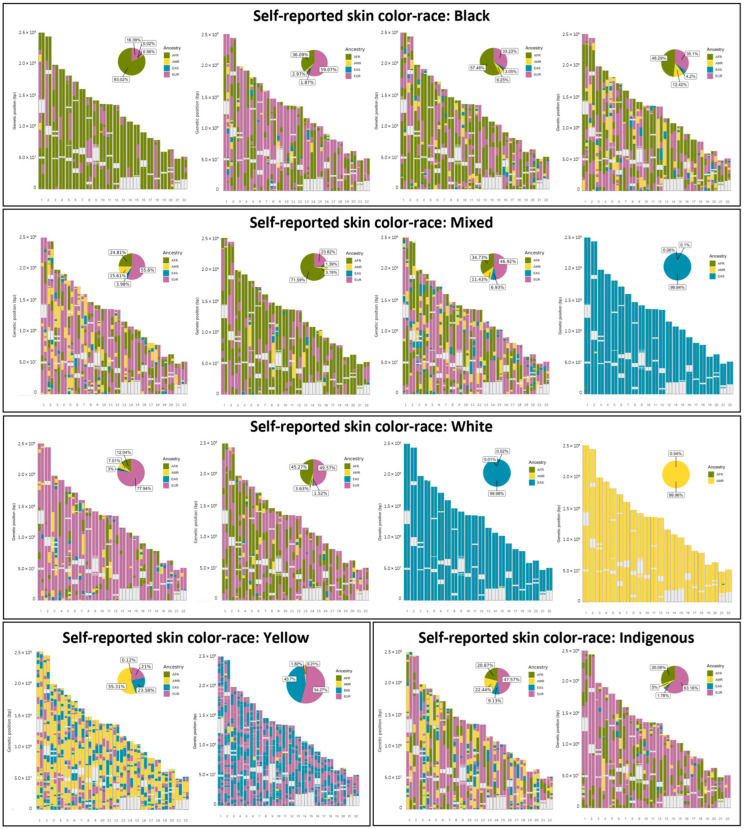
Examples of ideograms of individuals with different patterns of local ancestry according to self-reported skin color/race.

**Table 1 genes-15-00917-t001:** Characteristics of 2015 ISA-Nutrition population according to self-reported skin color/race for unrelated individuals (n = 720). São Paulo, 2015. Appendix A presents the distribution for the total ISA-Nutrition sample (n = 841).

	2015 ISA-Nutrition		Self-Reported Color/Race												*p*-Value
			Black		Indigenous		Mixed		White		Yellow		Not Answered		
	**n**	**%**	**n**	**%**	**n**	**%**	**n**	**%**	**n**	**%**	**n**	**%**	**n**	**%**	
**Unrelated Population**	720	100	74	10.3	2	0.3	261	36.3	364	50.6	10	1.4	9	1.3	
** *Age group* **															
**Adolescent (12–20 y)**	213	29.6	26	35.1	0	0	97	37.2	87	23.9	1	10.0	2	22.2	
**Adult (20–60 y)**	227	31.5	24	32.4	1	50.0	84	32.2	117	32.1	1	10.0	0	0	
**Older adult (60–94 y)**	280	38.9	24	32.4	1	50.0	80	30.7	160	44.0	8	80.0	7	77.8	0.002 ^a^
** *Sex* **															
**Female**	339	47.1	38	51.4	0	0	110	42.2	182	50.0	5	50.0	4	44.4	
**Male**	381	52.9	36	48.6	2	100	151	57.8	182	50.0	5	50.0	5	55.6	0.115 ^a^
***Ancestry Proportion* > 50%**															
**Sub-Saharan African**	61	8.5	42	56.8	0	0	18	6.9	0	0.0	0	0	1	11.1	<0.001 ^b^
**Native American**	20	2.8	0	0	0	0	4	1.5	6	1.70	10	100	0	0	0.806 ^b^
**European**	578	80.3	14	18.9	1	50.0	210	80.5	345	94.8	0	0	8	88.9	<0.001 ^a^
***Ancestry Proportion* > 0%**															
**Sub-Saharan African**	589	81.8	74	100	2	100	254	97.3	244	67.0	8	80	7	77.8	<0.001 ^b^
**Native American**	581	80.7	69	93.2	2	100	246	94.3	248	68.1	10	100	6	66.7	<0.001 ^a^
**European**	706	98.1	73	98.7	2	100	260	100	362	99.5	0	0	9	100	0.620 ^a^
***Ancestry Proportion* (%)**	**Median**	**IQR**	**Median**	**IQR**	**Median**	**IQR**	**Median**	**IQR**	**Median**	**IQR**	**Median**	**IQR**	**Median**	**IQR**	
**Sub-Saharan African**	18.2	4.1–32.2	56.1	41.9–72.7	23.8	19.4–28.2	26.5	19.0–36.4	7.4	0–17.6	3.4	3.3–3.7	10.1	1.7–11.0	0.0001 ^c^
**Native American**	6.10	1.3–11.2	5.6	2.7–9.9	23.4	6.9–39.9	8.5	4.8–12.7	3.60	0–8.9	96.7	96.3–96.7	1.3	0–9.3	0.0001 ^c^
**European**	71.5	55.6–89.4	35.8	21.7–48.3	52.7	40.6–64.9	62.3	53.4–72.3	86.3	72.5–1	0	0	80.0	75.0–98.1	0.0001 ^c^

IQR = interquartile range, ^a^ *p*-value for Chi squared test for Black, Mixed and White only. ^b^ *p*-value for Fisher’s exact test for Black, Mixed and White only. ^c^ Kruskal–Wallis equality of populations rank test for Black, Mixed and White only. Note: The p-values in the last column are based on tests where the null hypothesis is that there is no association between the variables and self-reported color/race.

**Table 2 genes-15-00917-t002:** Self-reported skin color/race as Black (N = 74), Mixed (N = 261), and White (N = 364) according to quartiles of individual Sub-Saharan African (AFR), Native American (AMR) and European (EUR) ancestry for the unrelated ISA sample. São Paulo, 2015.

	Quartiles of AFR Ancestry											
Self-Reported Skin color/race	1rst		2nd		3rd		4th		Median Regression Model		0.75 Regression Model	
	N	%	N	%	N	%	N	%	β (95%CI)	*p*-Value	β (95%CI)	*p*-Value
**Black**	0	0	2	2.7	3	4.1	69	93.2	ref		ref	
**Mixed**	19	7.30	46	17.6	105	40.2	91	34.9	−0.28 (−0.32; −0.24)	<0.001	−0.35 (−0.42; −0.29)	<0.001
**White**	168	46.2	119	32.7	58	15.9	19	5.2	−0.46 (−0.50; −0.42)	<0.001	−0.54 (−0.60; −0.48)	<0.001
									PseudoR^2^ = 0.35		PseudoR^2^ = 0.36	
	**Quartiles of AMR ancestry**											
	**1rst**		**2nd**		**3rd**		**4th**		**Median Regression Model**		**0.75 Regression Model**	
	**N**	**%**	**N**	**%**	**N**	**%**	**N**	**%**	**β (95%CI)**	***p*-value**	**β (95%CI)**	** *p* ** **-value**
**Black**	14	18.9	30	40.5	16	21.6	14	18.9	ref		ref	
**Mixed**	31	11.9	68	26.1	82	31.4	80	30.7	0.03 (0.01; 0.05)	0.008	0.02 (−0.01; 0.06)	0.186
**White**	153	42.0	83	22.8	64	17.6	64	17.6	−0.02 (−0.04; 0.00)	0.056	−0.01 (−0.04; 0.03)	0.646
									PseudoR^2^ = 0.07		PseudoR^2^ = 0.02	
	**Quartiles of EUR ancestry**											
	**1rst**		**2nd**		**3rd**		**4th**		**Median Regression Model**		**0.75 Regression Model**	
	**N**	**%**	**N**	**%**	**N**	**%**	**N**	**%**	**β (95%CI)**	***p*-value**	**β (95%CI)**	***p*-value**
**Black**	64	86.5	8	10.8	2	2.7	0	0	ref		ref	
**Mixed**	82	31.4	101	38.7	60	23.0	18	6.9	0.27 (0.22; 0.33)	<0.001	0.22 (0.18; 0.25)	<0.001
**White**	27	7.4	54	14.8	108	29.7	175	48.1	0.50 (0.45; 0.55)	<0.001	0.43 (0.39; 0.47)	<0.001
									PseudoR^2^ = 0.30		PseudoR^2^ = 0.34	

Quantile regression adjusted for age. β is the coefficient model, and 95%CI is the 95% confidence interval.

## Data Availability

The datasets generated and/or analyzed during the current study are available in the European Genome-phenome Archive (EGA) repository (https://ega-archive.org/ (accessed on 12 July 2024)), (study ID EGAS00001007818). The scripts used in global and local ancestry analysis for the manuscript are publicly available at the following: https://github.com/camila-alves/ancestry_isa (accessed on 12 July 2024).

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
