# Peer review of "Genetic Ancestry and Self-Reported “Skin Color/Race” in the Urban Admixed Population of São Paulo City, Brazil"

_genes, 2024, doi:10.3390/genes15070917_

Round 1
Reviewer 1 Report
Comments and Suggestions for Authors
In this research, the authors with an elevated dataset and a well-documented analyses, further demonstrate that even if genetic differences exist between different human groups, we cannot speak of ancestry or human races based on skin color.
in the introduction, the authors could be more critical of the terminology and scientific evidence using genetic markers that highlight that there are no biological races in humans.
Even if genetic differences between human groups are of fundamental importance at an epidemiological level and for gene therapies, skin color race self-reports cannot be used because they are subjective and very arbitrary. The authors proposed a genetic analyses and data analyses workflow that could be the basis of the genetic-based medical therapies.
The findings of this research emphasize the critical importance of accurately defining the terms race, ancestry and skin color and thoroughly analyzing them in population studies, particularly within admixed populations, as Brazilian populations, Latin American countries, but also in North African populations, Near-Eastern Asian or Eastern Mediterranean populations.
The authors should enfatize in the abstract the concept that as precision medicine evolves and access to molecular testing expands, to improve study quality and improve overall health outcomes for mixed populations, regardless of skin color, race, or ancestry, a pool of genetic markers should be analyzed to determine the genotype of each subject. In this case, evolutionary medicine or personalized medicine could prove to be a success.
Moreover, in my opinion it is better change all the terms regarding races and Skin Color-Race with geographical terms. For this reason, all over the text, the authors should change Sub-Saharan African with South Saharan, black with African origin or ancestry, yellow with Asian ancestry…..
In the keywords, the authors must eliminate the words already reported in the title of the manuscript and include others.
In figure 2 are not clear the different symbols and colours.
In figure 5 the numbers of the chromosomes are overlapping.
Reviewer 2 Report
Comments and Suggestions for Authors
Thank you for the opportunity to review the article, which was very interesting to read. The authors have done a lot of important and valuable work, including a detailed statistical analysis of a fairly decent sample, the article is well structured and organized. The article is well structured and organized. The work is very impressive, statistically robust, and has a consistent approach to design and discussion. I was particularly impressed by the quality of the communication of the research findings, which in most cases was clear and easy to read despite the technical complexity. Equally impressive is the consideration of the limitations of the research findings presented (within their important findings, of course). Despite the many positive aspects, I made a few minor recommendations that should be considered by the authors:
- In the introduction, where the authors characterize the terms "race", "ethnicity", and "ancestry", they should point out that in recent years the term “race“ is seen as a pejorative term and ethnic diversity is used instead.
- I understand that in the MM section you reference the earlier study where the participants were described, but I still think it's important to at least mention the exact age range (12 years and older is not enough) and biological or self-reported sex of the study participants, that should be in the MM section, not in the results.
- What exactly does "free-living healthy individuals" mean? First, it should be explained what the term "free-living" refers to, especially when it comes to sub-adults, and second, how was cardiometabolic syndrome determined if the individuals were healthy, which you mention?
- In subsection 2.4, you should indicate which statistical program you used.
- What use do the results of the study have for clinical practice, i.e. for further epidemiologic studies? I suspect that any further study that takes ethnicity into account will rely on self-reporting of ethnicity, as genetic analysis cannot always be done, e.g. because it is too time consuming or too expensive. How should this be handled then? For example, when it comes to your earlier study by Fisberg et al. 2015. Or with all other population studies in general. So how can it be applied in clinical practice or epidemiologic studies? What outcomes can it influence – perhaps to mention some diseases for which it would be particularly relevant, for example in mixed populations? I suggest that these points should be emphasized and discussed in more detail.
- There is no need to refer to the figures in the discussion as they have already been interpreted in the results section.
- Figure 5 – the numbers on the x-axis should be revised as they are not easy to read
- In Figure 2, please use the abbreviation CP1/2 in the legend, e.g. where it says: principal component (CP1, 2)
- I suggest revising Figure 6, as it is difficult to read without zooming, and paying more attention to what the individual graphs represent in relation to the self-reported skin color. For example, why is there only one for "yellow"?
- The supplementary Table 2 is not sufficiently legible.
- The legend in Figure 3 should be slightly revised to make it easier for the reader to understand. Please add the information that the numbers in each region represent the number (?or percentage) of individuals. What then can the reader understand by “other countries” and the same goes for missing – missing what?
- There seem to be two conflicting statements in the conclusion: First, you emphasize that the results of your study show that it is important to define the terms studied and to distinguish between them in population studies, especially in mixed studies, but at the end of the conclusion you state that the results of your study have the potential to increase the quality of studies and improve overall health outcomes for populations regardless of skin color- race, and ancestry. Based on the recommendations above, please explain this in a way that does not sound confusing to the reader.
Comments on the Quality of English LanguageMinor editing of English language required
